# Optimization of Different Acid-Catalyzed Pretreatments on Co-Production of Xylooligosaccharides and Glucose from Sorghum Stalk

**DOI:** 10.3390/polym14040830

**Published:** 2022-02-21

**Authors:** Xiaocui Yang, Xiaoliu Liu, Yequan Sheng, Hanzhou Yang, Xinshuai Xu, Yuheng Tao, Minglong Zhang

**Affiliations:** 1Engineering Training Center, Nanjing Vocational University of Industry Technology, Nanjing 210023, China; 2019010152@niit.edu.cn; 2College of Bioscience and Engineering, Hebei University of Economics and Business, Shijiazhuang 050061, China; liuxiaoliu@heuet.edu.cn; 3Jiangsu Co-Innovation Center of Efficient Processing and Utilization of Forest Resources, International Innovation Center for Forest Chemicals and Materials, College of Materials Science and Engineering, Nanjing Forestry University, Nanjing 210037, China; 1352124091@njfu.edu.cn (H.Y.); limimingjun@163.com (X.X.); 4Department of Bioengineering, School of Pharmacy & School of Medicine, Changzhou University, Changzhou 213164, China; cztyh0305@163.com; 5College of Materials Science and Engineering, Central South University of Forestry and Technology, Changsha 410004, China; 6Anhui Hongsen Hi-Tech Forestry Co., Ltd., Bozhou 233600, China

**Keywords:** sorghum stalk, xylooligosaccharide, acid hydrolysis, enzymatic hydrolysis, glucose

## Abstract

There is an increasing emphasis on the transformation of lignocellulosic biomass into versatile products. The feasibility of preparing xylooligosaccharides (XOS) by hydrolysis of sorghum stalk (SS) using organic and inorganic acids was studied. The influences of different acids (gluconic acid, acetic acid, sulfuric acid, and oxalic acid), process time and temperature on the hydrolysis of SS were explored. The findings indicated XOS yield can be maintained at a high level under different conditions with organic acid pretreatments. Optimum yield of XOS (39.4%) was obtained using sulfuric acid (pH 2.2) at 170 °C and 75 min of process time. It is suggested when reaction temperature and time were increased, both X5 and X6 are cracked into XOS with lower molecular mass such as X2, X3, and X4. Moreover, the results based on mass balance showed that up to 110 g (XOS) plus 117 g (glucose) can be recovered from 1000 g of SS. Results will give insights into establishing an efficient acid pretreatment of sorghum stalk to coproduction of XOS and glucose.

## 1. Introduction

Consistent efforts have been adopted using renewable lignocellulosic material as feedstock to produce value-added products in the biorefinery industry, such as biofuel, biomaterial [1,2,3]. Lignocellulosic material represents the primary constituents of biomass that comprising mainly the complex polymeric compounds of lignin, multiple cross-linked structures of cellulose and hemicellulose [4]. It is generally known that both hemicellulose and cellulose can be hydrolyzed by enzyme or chemical treatment into glucose and xylose for subsequent conversion into valuable feedstocks for instance xylonic acid and ethanol [5,6].

However, due to the presence of crystalline and high polymerization degree of cellulose that embedded/connected to the lignin and hemicellulose, the overall lignocellulosic material shows high resistance to enzyme and chemical treatment [7].

Sorghum stalk (SS) (a residue of the sorghum production) accounts for about 35% of the total sorghum mass. About 1.4 million tons of SS is generated in China every year, more than half of those SS are burned directly without proper reusing strategy. Since SS is found with 30–45% cellulose and 25–30% xylan-type hemicellulose [8], hence it is hypothesized that the SS could be an encouraging material to refine versatile materials such as fermentable sugars and XOS [9] rather than recovering no value from burning.

Hemicellulose is the second abundant lignocellulosic material in nature with various degree of polymerization [10]. Xylooligosaccharides (XOS) is the product resulted from hydrolysis of xylan-type hemicellulose that has lower degree of polymerization. This includes xylobiose (X2), xylotriose (X3), xylotetraose (X4), xylopentaose (X5), and xylohexaose (X6) [11,12]. At present, XOS is often used as a functional ingredient to stimulate the growth of prebiotic activity in the intestine for improvement of immunity. In addition, XOS could also be applied in food, animal feed and pharmaceutical fields as well as to reduce cholesterol levels and the risk of colon cancer [13].

Suitable pretreatment approach represents a crucial step to resolve the cellulose entanglement for effective utilization of lignin, hemicellulose and cellulose. Lately, the widely explored pretreatment methods of lignocellulose materials include steam explosion, acids-based thermal pretreatment, solvent extraction and biological pretreatment [14,15]. Acid pretreatment is an effective way to ease the conversion of cellulosic biomass into biochemical [16]. It is a commonly used chemical pretreatment technology to remove cellulose by degrading it into oligosaccharides. Acid pretreatment is considerably low cost and suitable for the production of biochemical [17,18]. During acid pretreatment reaction, the glycosidic bonds between the hemicellulose will be broken which subsequently facilitates the degradation of cellulose into oligosaccharides and monosaccharides in the later enzymatic hydrolysis [19]. However, there has not been detailed research on the production of biochemicals from SS using various acid with different strength.

Organic acids have lower pKa values and thus provides milder acid condition than mineral acids, such as sulfuric acid. As an organic acid, they may impart a more readily tuned, optimized, and efficient hydrolysis of various lignocellulosic substrates over a range of temperatures and times compared to stronger mineral acids. Therefore, in this research, different types of acid (i.e., oxalic acid, sulfuric acid, gluconic acid, acetic acid) pretreatments was applied to produce XOS and glucose from SS. The influences of different acids, the process time and temperature to perform hydrolysis on the composition and yield of XOS were investigated. Subsequently, the ability of enzymatic hydrolysis of residual cellulose were systematically analyzed. Further, the mass balance diagrams for the overall process is presented to demonstrate the effectiveness. It was observed the great potential of using acid pretreatment on SS to produce value-added biorefinery products.

## 2. Materials and Methods

### 2.1. Materials

Sorghum stalk (Sorghum bicolor var. kaoliang) (SS) was collected from a farmland in Shandong Province, China, directly after crop harvest. After air-drying, the stalks were chipped using an industry-scale Drum-Chipper. The chips were then reduced into powders using a laboratory-scale grinding miller. These SS powders were sieved using a 40-mesh sieve followed by mild drying at 60 °C in an electric oven for 1 day, the dried SS contained 4.2% moisture. Sulfuric acid (SA), oxalic acid (OA), gluconic acid (GA), acetic acid (AA) were purchased from Macklin, China. The pKa values of sulfuric acid, oxalic acid, gluconic acid and acetic acid were −2.0, 1.27, 3.86 and 4.74, respectively. Distilled water was used as reaction medium whenever necessary.

### 2.2. Evaluation of Acid Pretreatment

The acid pretreatment process was assessed via CSF (also known as combined severity factor), which consolidates the data of various combinations of process time (min) and temperature (°C) used to perform hydrolysis.

The CSF was calculated as following [20]:$$\begin{array}{c} \mathrm{CSF}=\log\left(R_{0}\times [H^{+}\right])=\log R_{0}-pH\\ R_{0}=t\times \exp\left[\left(T_{h}-T_{r}\right)/w\right] \end{array}$$
where T*_h_* and T*_r_* refer to the hydrolysis temperature and base temperature (100 °C), respectively. w refers to fitting parameter constant (14.75). The CSF has been specifically explained as the influence of acid pretreatment in the materials. It originated in the use of acidic pretreatment on biomass to estimate the solubilization of xylan and the reduction of lignin. The combined index was then used to further assess the influences of hydrolysis performed via various acids (SA, OA, GA and AA). The reaction pH in different acid pretreatments were uniformly adjusted to 2.2. The biomass to acid liquid ratio was 1:10. The combinations of reaction conditions were tabulated in Table 1 based on the previous studies [19,21].

### 2.3. Enzymatic Hydrolysis

After acid pretreatments, the pretreated SS were hydrolyzed by cellulase (C2730) manufactured by Novozymes, Sigma Co. in Shanghai, China. Prior for enzymatic hydrolysis, the acid pretreated SS was constantly cleaned using distilled water for 6 h to eliminate the remaining acids and toxic aldehydes. Then, enzymatic hydrolysis of the cleaned acid pretreated SS was performed in a 25 mL screw-capped bottle and mixed with a dose of 20 FPIU/g. The process was conducted for 108 h under a temperature of 50 °C and mild acidic pH (4.8) with constant shaking (150 rpm) in a water bath. During enzymatic hydrolysis, the contents of glucose and cellobiose were quantified at 12-h intervals in the enzymolysis solution.

### 2.4. Analytical Methods

The contents of lignin, hemicellulose and cellulose in the raw material and acid pretreated SS were resolved by the National Renewable Energy Laboratory method (NREL) [22]. The xylobiose (X2), xylotriose (X3), xylotetraose (X4), xylopentaose (X5), and xylohexaose (X6) were quantified by high performance anion exchange chromatography-pulsed amperometric detector at 30 °C (HPAEC-PAD, Dionex ICS-3000) with a CarboPacTM PA200 column using NaOH (100 mM) and NaOAc (500 mM) under constant flow rate of 0.3 mL/min [23]. The contents of glucose, xylose and arabinose were quantified by high performance liquid chromatography at 55 °C (HPLC, Agilent 1260) with analytical column (Aminex Bio-Rad HPX-87H) adopting H2SO4 (5 mM) at a fixed flow rate of 0.6 mL/min. All samples were physically refined through a 0.22 μm membrane to eliminate solid impurities before analysis. Each experiment was repeated twice.

The yield of XOS and enzymatic hydrolysis were determined as follows:(1)$$\mathrm{XOS}\;\mathrm{yield}\;\left(\%\right)=\frac{\left(X2+X3+X4+X4+X6\right)\;\left(g\right)\times 100}{\mathrm{Initial}\;\mathrm{xylan}-\mathrm{based}\;\mathrm{hemicellulose}\;\left(g\right)}$$
$$\mathrm{Enzymatic}\;\mathrm{hydrolysis}\;\mathrm{yield}\;\left(\%\right)=\frac{\left(\mathrm{cellobiose}+\mathrm{glucose}\right)\times 0.9\;\left(g\right)\times 100}{\mathrm{Initial}\;\mathrm{glucan}\;\mathrm{content}\;\mathrm{of}\;\mathrm{materials}\;\left(g\right)}$$

## 3. Results and Discussion

### 3.1. XOS Yield by Four Kinds of Acid Pretreatments of Sorghum Stalk

It is generally known that the lower dissociation constant of an acid, the greater of its ability to release H+, hence causing more degradation of hemicellulose to XOS and monosaccharides. However, when comparing with strong organic acids at the same pH value, the concentrated AA could provide more stable and sufficient H+ since AA is a weak electrolyte and hence lower dissociation balance (Table 1). Consequently, the presence of concentrated H+ could enhance the rupture effects in pre-treatment of SS, and degrade the macromolecule xylan-based hemicellulose into smaller molecule of oligosaccharides [24]. The results showed that the weaker acid was conducive to the further degradation of XOS under the same pH condition.

### 3.2. Influences of the Process Temperature and Time

The influences of process temperature and time on the yield and components of XOS in the pretreatment of SS were shown in Figure 1. The highest XOS yield (38.6%) was obtained from the pretreatment of AA for 60 min at 152.5 °C with a constant pH (Figure 1). With an increase of temperature, XOS yield first increased and then decreased at 152.5 °C. While a sharp decrease in XOS yields occurred at reaction time (60 min) from 18.3% (170 °C) to 0.1% (180 °C). This indicates that hydrolysis performed under longer time plus high temperature conditions were unfavorable for XOS production. On the other hand, the XOS yield improved from 2.0% (127.8 °C) to 38.6% (152.5 °C) at the same reaction time (60 min). This could be attributed to the use of relatively higher temperature accelerated the hydrolysis of xylan-type hemicellulose to low-molecular oligosaccharides. Noureddini and Byun [25] also reported the same phenomenon. In addition, the same findings were produced with OA and GA in which the XOS yields of were 27.5% (170 °C, 75 min) and 32.1% (152.5 °C, 60 min), respectively.

Moreover, Figure 1 shows the XOS yield obtained from the hydrolysis of SS with AA and GA. As the time progress, the yield of XOS decreased. In comparison, the yield of XOS increased with longer reaction time using SA and OA. This indicates that the yield of XOS obtained from the degradation of hemicellulose was significantly affected by different acids used. In addition, it was noticed from the hydrolysis with various acids that higher temperature is required to optimize the yield of XOS when strong acid is used [19].

Figure 2 presents the relative content of X2, X3, X4, X5, X6 produced by SS under different reaction temperature and time. When the reaction temperature is increased, both X5 and X6 are cracked into XOS with lower molecular mass such as X2, X3 and X4. Same observations were obtained at different reaction time. The results showed that both process parameters (i.e., temperature and time) have noticeably influences on the molecular mass distribution of the resulted products. Figure 2a,b demonstrate the influence of different reaction time on the yield of X2–X6. When the hydrolysis undergone longer, both X2 and X3 increased whereas for X5, X6 and >X6, the hydrolysis process was continued until producing oligosaccharides with lower molecular mass such as X2, X3 and X4. It can be clearly observed that a relatively low yield of X5 and X6 was obtained at high temperature (170 °C, 75 min) with organic acid pretreatment. Although the higher yield of XOS could be attained by the use of concentrated acid or prolong process time, the unwanted side products such as furfural, hydroxymethyl furfural and other harmful substances might be resulted [26].

### 3.3. Enzymatic Hydrolysis of Sorghum Stalks for Fermentable Sugar Production

When the maximum XOS was obtained by hydrolysis of SS with different acid pretreatment, the chemical properties of the pretreated solid fraction was identified (Table 2). In favor of optimizing the yield of the pretreated materials, the residues pretreated with different acids were mixed with 20 FPIU/g of glucan to perform the enzymatic hydrolysis under process time of 108 h, the results are displayed in Figure 3. Among them, the enzymatic hydrolysis yield obtained from SS pretreated with OA and SA was the highest (exceeding 42%). Zhang et al. [27] obtained 43.6% yield from the combination of supercritical CO2 and ultrasonic pretreatment on SS. In this research, approximately 33% yield were obtained from the enzymatic hydrolysis of AA and GA pretreated SS, while the XOS yields produced was 38.6% and 32.1%, respectively, indicating that both AA and GA pretreatments fully removed the hemicellulose in the SS.

### 3.4. Mass Balance for Production of XOS and Glucose

Figure 4 and Appendix A illustrate the analysis of mass balance to produce XOS and glucose. About 1000 g of dried SS containing 336 g of glucan, 274 g of xylan, and 235 g of lignin. Up to 110.2 g of XOS and 619.0 g of solid fraction (containing 268 g glucan, 94 g xylan, 151 g lignin) were produced from SS treated with SA (pH = 2.2) for 75 min at 170 °C (Figure 4). When comparing with the SA hydrolysis used in Figure 4, the two-stage process of organic acid pretreatment and the enzymatic hydrolysis produced lower content of XOS and fermentative sugars (Appendix A). In addition, XOS uses high-precision high performance anion exchange chromatography-pulsed amperometric detector (HPAEC-PAD) for analysis in this study. These results indicated that pretreatment of SS with SA in optimum condition was beneficial for XOS and fermentable sugars co-production.

## 4. Conclusions

In this work, the feasibility of preparing XOS and glucose from SS by different acid pretreatments were explored. It was found that the hydrolysis conditions such as types of acid, process time and temperature significantly influenced the XOS yield. The pretreatment using sulfuric acid (pH 2.2) produced an optimum yield of XOS (39.2%) at the process conditions of 170 °C and 75 min. However, organic acid pretreatments were easier to get high content of XOS under several conditions compared to sulfuric acid pretreatment. Results also indicated that higher process time and temperature caused X5 and X6 to split into lower molecular mass such as X2, X3, and X4. An approximately of 110.2 g XOS and 117 g glucose were produced from 1000 g of SS based on mass balance during sulfuric acid pretreatment. To conclude, acid pretreatments exerted promising effects for facilitating coproduction of XOS and glucose. In the future work, the effects of lignin in SS on the depolymerization of xylan and the production of xylose oligosaccharides during different acid pretreatments would be explored.

## Figures and Tables

**Figure 1 polymers-14-00830-f001:**
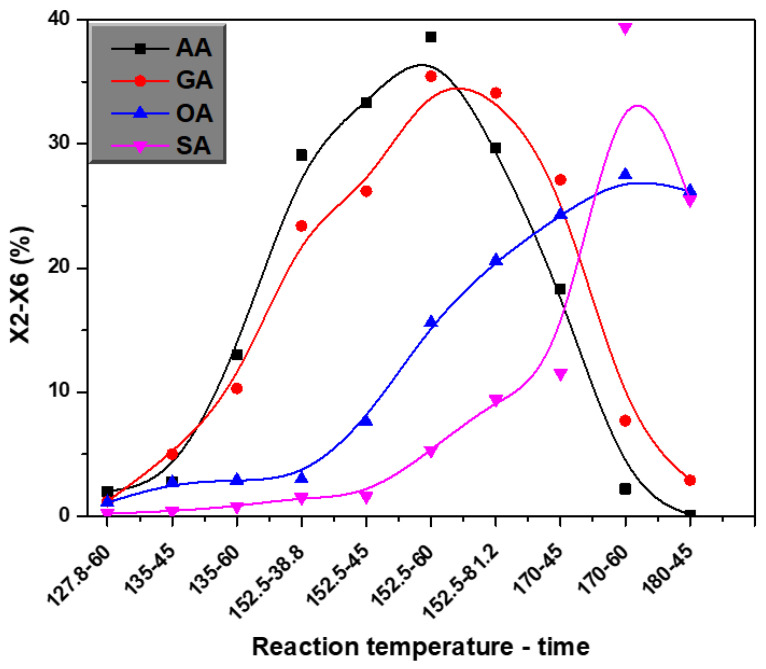
Influence of reaction temperature and time on xylooligosaccharides (XOS) yield.

**Figure 2 polymers-14-00830-f002:**
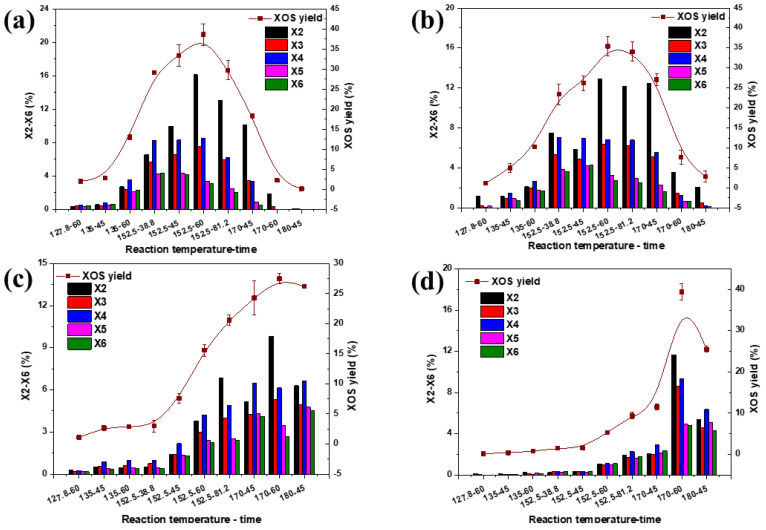
The yields of X2–X6 produced from sorghum stalks with different reaction temperature and time under different acids ((**a**) Acetic acid, (**b**) Gluconic acid, (**c**) Oxalic acid, (**d**) Sulfuric acid).

**Figure 3 polymers-14-00830-f003:**
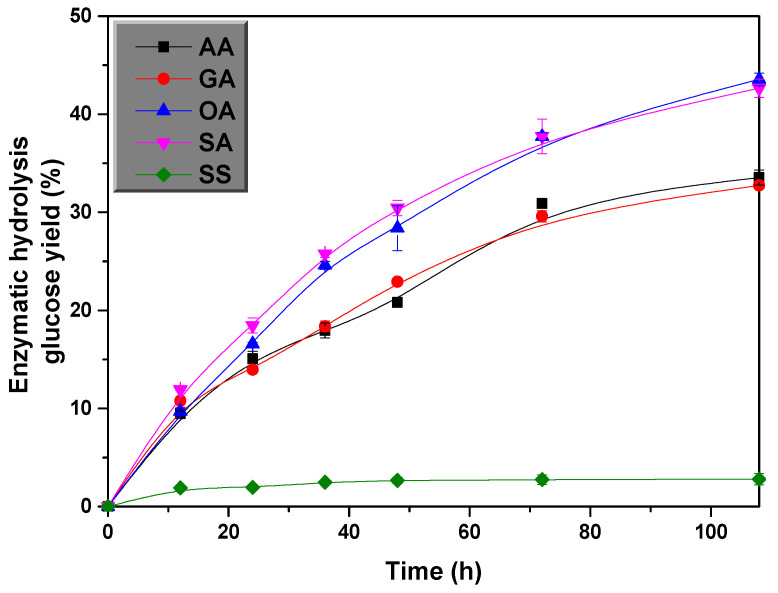
The 108 h enzymatic hydrolysis efficiency of SS subjected to different acids hydrolysis.

**Figure 4 polymers-14-00830-f004:**
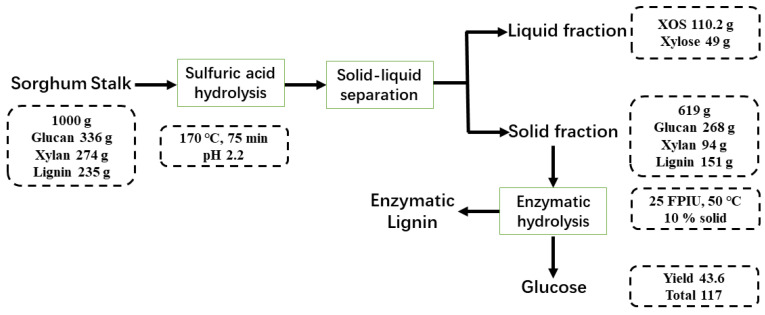
Mass balance of the co-production of XOS and glucose from SS.

**Table 1 polymers-14-00830-t001:** The xylooligosaccharides (XOS) yield of response surface analysis.

Reaction Acids/pH (2.2)	Run	Variable		Response
		Temperature (°C)	Time (min)	XOS Yield (%)
Acetic acid (AA)	1	127.8	60	2.0
	2	135	45	2.8
	3	135	75	13.0
	4	152.5	38.8	29.1
	5	152.5	45	33.2
	6	152.5	60	38.6
	7	152.5	81.2	29.7
	8	170	45	18.3
	9	170	75	2.2
	10	180	45	0.1
Gluconic acid (GA)	1	127.8	60	1.2
	2	135	45	5.2
	3	135	75	10.3
	4	152.5	38.8	23.4
	5	152.5	45	26.2
	6	152.5	60	32.1
	7	152.5	81.2	30.7
	8	170	45	27.1
	9	170	75	7.7
	10	180	45	2.9
Oxalic acid (OA)	1	127.8	60	1.1
	2	135	45	2.7
	3	135	75	2.9
	4	152.5	38.8	3.0
	5	152.5	45	7.6
	6	152.5	60	15.6
	7	152.5	81.2	20.6
	8	170	45	24.3
	9	170	75	27.5
	10	180	45	26.2
Sulfuric acid (SA)	1	127.8	60	0.2
	2	135	45	0.4
	3	135	75	0.8
	4	152.5	38.8	1.5
	5	152.5	45	1.6
	6	152.5	60	5.3
	7	152.5	81.2	9.4
	8	170	45	11.5
	9	170	75	39.4
	10	180	45	25.8

**Table 2 polymers-14-00830-t002:** Chemical composition of the solid fraction after acids treatment.

Samples	Cellulose (wt%)	Xylan (wt%)	Araban (wt%)	Lignin (wt%)
SS	33.6	27.4	2.7	23.5
AA	44.4	15.6	/	25.2
GA	37.4	12.3	/	25.4
OA	43.3	14.5	/	24.9
SA	43.3	15.2	/	24.5

## Data Availability

The data presented in this study are available on request from the corresponding author.

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
