# Peer review of "Optimization of Different Acid-Catalyzed Pretreatments on Co-Production of Xylooligosaccharides and Glucose from Sorghum Stalk"

_polymers, 2022, doi:10.3390/polym14040830_

Round 1
Reviewer 1 Report
Manuscript is well written however there are some areas need to improve. Please elaborate the combinations of reaction conditions (Table 1). Is there any background data or study? Authors claimed cost-effective process for preparation of XOS. There should be a section in the manuscript with in-dept cost assessment of the process. Please improve the figure quality.
Reviewer 2 Report
The authors study the effect of different types of acid on the synthesis of the xylooligosaccharides. It is an important study to understand the effect of PH, temperature and process time on the yield of the XOS. However, there are mistakes found in the articles and I suggest the following revisions to be implemented in the manuscript.
1. Avoid using “our” in writing the article.
2. The objective of the study is unclear. It would be better for the author to highlight the state of the art for the current study and the gap from the previous literature.
3. increased sharply from 18.3% to 0.1% when the temperature increased from 170 oC to 180 o Should be 170 oC.
4. It would be better for the author to highlight the future work follow by current study.
Reviewer 3 Report
This document shows interesting and innovative research, aiming at the production of oligosaccharides. The manuscript has major issues that must be fixed. I have a few comments to give to the authors, and I hope that they can be helpful
**Title: I understand that it is a comparison, but I think “Analyses” or “Optimization” sounds better than “comparison”.
**Abstract: The conclusions seem to not be related to the results. I recommend the authors rethink the last part of the abstract.
**Keywords: OK
**Introduction:
-The current flow is:
Use of natural feedstocks for value-added products and structure of lignocellulosic materials and need to pretreat lignocellulosic materials to extract the biopolymers > hemicellulose and xylooligosaccharides > sorghum stalk > acid pretreatment and originality of this work > objective of this work.
The flow must be improved to clarify the aim of this work for the readers. I recommend the following flow:
Use of natural feedstocks for the bioeconomy > structure of the lignocellulosic materials > sorghum stalk > hemicellulose and xylooligosaccharides > need to pretreat lignocellulosic materials to extract the biopolymers > acid pretreatments > objective and originality of this work
Please, keep in mind that there is extensive literature using sorghum stalks to extract glucose and use it for bioethanol fermentation. The novelty of this work is the extraction of the xylooligosaccharides. Please, keep the focus on these xylooligosaccharides, not glucose. The glucose is a bonus, but the most important thing of this document is the xylooligosaccharides.
-Please, avoid the use of “good bacteria” in a scientific manuscript. This is useful for a general audience, but for a scientific document, I recommend writing “prebiotic activity”.
**Materials and Methods
*Materials
-The authors should provide more information about the sorghum stalk. Were the stalks collected after the harvest? What is the variety? Where were the plants produced? Were they pre-processed before sieving? Didn’t the authors chop or grind the stalks in a mill? All of these details and others are important because this is a natural material, with natural variation, and if other researchers replicate this work in the future, they must know the details about the farm, not only the laboratory.
-How did the authors know the lignocellulosic composition? If they analyzed the raw material, the results should be presented in the Results and Discussion section, not the Materials and Methods section.
*Evaluation of acid pretreatment
-Please, provide the reference for the equation and CSF.
-How did the authors perform the hydrolysis? If they used a commercial reactor, they must provide the maker and model. If they used a custom-made reactor, they must provide a technical description and a picture of it, with an explanation of how it works.
-It is also important to provide detail about the ratio biomass: acid solution. Did the authors use the same ratio for the four treatments, let us say 1 g of biomass for 20 mL of the acid solution, or did they use different ratios for different mixtures?
*Statistical analysis
-The authors did not perform statistical analysis in the experiment. If they repeated the experiment twice, they can make an analysis. I recommend contacting a statistician and discussing a few ways to analyze the graphs, evaluating the joint impact of temperature and time.
**Results and Discussion
-Line 140: typo: 170°C, not 170 oC. Additionally, do not keep a space between the value and the “°C” wherever the authors are mentioning temperature.
-Lines 185-186: Please, draw conclusions only based on the experimental results. The authors did not perform XRD analysis, so they cannot discuss crystallinity. They can hypothesize about this topic, but it is unrelated to the main objective of this work: which treatment is the best approach to extract XOS?
*Influences of the process temperature and time
-Figure 1: this figure is not a good way to represent the data. Figure 1a seems like the authors fixed a time and observed the impact of temperature, and figure 1b seems like the authors fixed a temperature and observed the impact of the time. Analyzing the data this way loses the overview of the situation and may mislead the authors. For example, it is not possible to verify the highest yields of each pretreatment (38.6% for AA, 32.1% for GA, 27.5% for OA, and 39.4% for SA). The authors must analyze the impact of temperature and time together, not isolating one from the other.
-Figure 2: This is an interesting figure, showing that the quality profile is impacted by the experimental conditions. However, the authors fail to explain if this quality impact is relevant. For example, for bacteria X, is it better X2 or X3 or X4 or a combination of 1:2:1 of X2:X3:X4, and so on? I strongly recommend the authors to just present the data that the experimental conditions impact the quality of XOS and discuss that more research is needed to detect the relevance of this for a given application.
*Mass balance
-This subsection is based only on the mass balance for sulfuric acid. If the authors want to make a mass balance analysis, they must calculate the balance for all the fours pretreatments, perform simulations based on the average and deviation values, like a bootstrapping, and analyze the simulated values. This section is misleading and, with no reasoning based on the results, the authors simply claim that “…SA pretreatment of SS represents a cost-effective method to produce XOS.”.
-Without a cost analysis, the authors cannot claim anything about cost-effectiveness. Even if a cost analysis is performed, it may reach different conclusions for different applications, and the authors will need to set the limits of this analysis. Furthermore, the authors must remember that the energy costs must be accounted for this type of analysis. Increasing the temperature from 152.5°C to 170°C and the time from 60 min to 75 min to change from the optimized conditions for acetic acid to the optimized conditions for sulfuric acid to increase 0.8% may not be economically attractive in certain situations.
*Conclusions
-The conclusions are not based on the experimental results. The results indicate that different experimental conditions, like acid, time, and temperature impact the yield and quality of XOS extracted from sorghum stalk. I strongly disagree with the text from lines 207 to 212 because depending on the application or life cycle analysis, hydrolysis with acetic acid or with sulfuric acid may be both adequate, or one can be more adequate. Sometimes, there is not only one “right answer”.
*Funding: OK
*Conflict of interest: OK
*Acknowledgment: Do not the authors want to acknowledge anybody? If you forgot to mention somebody, please, add this subsection. Polymers accept that the manuscripts have an Acknowledgment subsection at the “Back Matter” part (https://www.mdpi.com/journal/polymers/instructions).
*References:
-Please, avoid capitalizing all the nouns of a title even if the document was published this way. Therefore, for example, instead of writing “Opportunities for New Biorefinery Products from Ethiopian Ginning Industry By-products: Current Status and Prospects”, please write “Opportunities for new biorefinery products from Ethiopian ginning industry by-products: current status and prospects”.
Round 2
Reviewer 3 Report
The overall quality of the manuscript improved a lot and there are just small points that must be fixed:
**Title: OK
**Abstract: OK
**Keywords: OK
**Introduction:
-The current flow is:
Plant biomass and biomacromolecules > Sorghum stalk as a potential biomass feedstock > Hemicellulose and xylooligosaccharides > Pretreatments for biomass > Objective of this research
The flow is not bad and I think it should be kept as it is.
-Line 49: I am not sure, but shouldn’t be a “most” in line 49, making the sentence “Hemicellulose is the second most abundant…”?
**Material and methods
-Please write “Sorghum bicolor” in italics and keep “var. kaoliang” not in italics to follow the taxonomic rules.
**Results and discussions: OK
**Conclusions: OK
*Funding: OK
*Conflict of interest: OK
*Acknowledgment: OK
*References: OK